# A Supported Online Resilience-Enhancing Intervention for Pregnant Women: A Non-Randomized Pilot Study

**DOI:** 10.3390/ijerph21020209

**Published:** 2024-02-10

**Authors:** Sarah Van Haeken, Marijke Anne Katrien Alberta Braeken, Anne Groenen, Annick Bogaerts

**Affiliations:** 1Research & Expertise, Expertise Centre Resilient People, University Colleges Leuven-Limburg (UCLL), 3590 Diepenbeek, Belgium; anne.groenen@ucll.be; 2REALIFE Research Group, Faculty of Medicine, Department of Development & Regeneration, Women & Child KU Leuven, 3000 Leuven, Belgium; annick.bogaerts@kuleuven.be; 3REVAL-Rehabilitation Research Center, Faculty of Rehabilitation Sciences, Hasselt University, 3590 Diepenbeek, Belgium; marijke.braeken@uhasselt.be; 4Leuven Institute of Criminology (LINC), Katholieke Universiteit Leuven (KU Leuven), 3000 Leuven, Belgium; 5Faculty of Health, University of Plymouth, Devon PL4 8AA, UK

**Keywords:** resilience, pregnancy, postpartum, perinatal, maternal mental health, COVID-19, online intervention

## Abstract

A 28-week supported online intervention for pregnant women, informed by the Behavior Change Wheel Framework, was developed. The intervention included exercises, group sessions and a peer support platform. The aim of this study was to examine the potential effectiveness of the intervention in enhancing resilience and promoting maternal mental health. Using a quasi-experimental design, assessments were conducted at baseline, postintervention and follow-ups at six and 12 months after childbirth. Resilience, resilience attributes, and maternal mental health were measured using standardised scales. The intervention group received the intervention (*N* = 70), while the control group (*N* = 32) received care-as-usual. A repeated-measures ANOVA was used to determine within- and between-group changes. Results showed no significant differences between groups regarding resilience and maternal mental health. However, the intervention group demonstrated stable resilience (*p* = 0.320) compared to a significant decrease in the control group (*p* = 0.004). Within the intervention group, perceived social support remained stable during the intervention, but decreased significantly at the first follow-up (*p* = 0.012). All participants faced additional stress from the COVID-19 pandemic alongside the challenges of parenthood. This study contributes to maternal mental health literature with an innovative, supported online intervention. The intervention consists of different deployable components, designed to be offered online, and the current pilot data are promising. Further research is warranted to explore its full potential in clinical practice.

## 1. Introduction

The perinatal period, from conception to one year after childbirth, entails significant physical, psychological and social challenges [1]. Although many parents adapt well to the changes and challenges that they face, the perinatal period is an important time where parents are at increased risk of developing mental health problems [2]. Prevalence rates of maternal mental health problems (MMHPs) are around 20% [3,4], and are associated with adverse obstetric outcomes (e.g., preterm birth) [5,6,7]. Furthermore, offspring exposed to maternal distress in utero show an increased risk of developmental and mental health problems during childhood, adolescence and adulthood [8,9].

Studies regarding treatment interventions for MMHPs showed evidence for cognitive behavioural therapy (CBT) and interpersonal psychotherapy (IPT) [10]. The effect sizes for CBT and IPT approaches were larger in populations with a diagnosed psychiatric disorder, mainly perinatal depression [11]. Studies directed to non-clinical populations with possible subclinical symptoms suffered from low adherence rates and high demands on time and costs [12]. A possible explanation may be that these interventions might be too intensive, time-consuming or be experienced as stigmatising for women with no or subclinical symptoms [13].

Therefore, preventive interventions targeting pregnant women may be beneficial in reducing the risk of developing MMHPs. A recent review and meta-analysis of Waqas et al. [14] including 21 studies (12 randomized controlled trials (RCTs), 6 pilot RCTs, 2 quasi-experimental studies and 1 cluster RCT) investigated non-pharmacological preventive interventions for perinatal anxiety and depression. Specifically, psychosocial and psychological interventions have been effective in reducing the risk of developing MMHPs [14]. However, none of these interventions directly assessed or tried to enhance resilience.

To address this gap, a 28-week, supported, online resilience-enhancing intervention for pregnant women was developed. Resilience is known as an important protective factor against stress and the development of common mental health problems [15]. In general, resilience is defined as the ability to cope with challenges, stress and adversities in life [16]. Within the perinatal context, resilience is studied as a multi-factorial construct influenced by individual, socio-cultural and environmental factors [17]. We conducted a concept analysis and two-round Delphi survey on perinatal resilience, which defined resilience as: “*a circular process towards a greater wellbeing in the form of personal growth, family balance, adaptation or acceptance when faced with stressors, challenges, or adversity during the perinatal period*” (p. 11). Five main attributes of perinatal resilience were identified: social support, sense of mastery, self-efficacy, self-esteem, and personality [18].

The intervention development process was based on this conceptual model of perinatal resilience [18] and informed by the Behaviour Change Wheel (BCW) framework [19]. The intervention consisted of resilience-enhancing exercises, three online group sessions and an online peer support platform. An online method of delivery was chosen, offering the advantages of accessibility, flexibility and reduced stigma. Women were not required to attend time-consuming face-to-face sessions and could more easily combine their participation with their daily activities. Online interventions offer a certain degree of anonymity which might help women overcome the stigma regarding perinatal mental health problems [20]. The process of intervention development is described extensively elsewhere [21].

This study enrolled during the outbreak of the COVID-19 pandemic, where professional support became more critical but at the same time less accessible [22]. Building resilience is an important element of mental health promotion interventions for pregnant women, especially in such crises as the COVID-19 pandemic [23,24]. By integrating a resilience-oriented approach into prevention strategies, we expect to contribute to the wellbeing of pregnant women and their families.

The aim of this pilot study is to examine the potential effectiveness of the developed intervention for pregnant women in enhancing resilience and promoting maternal mental health. The present study explored the changes in resilience, resilience attributes and maternal mental health from pregnancy up to 12 months after childbirth among women who received the intervention and those who received care-as-usual. We will study within- and between-group differences.

## 2. Materials and Methods

### 2.1. Design and Sample Procedure

A quasi-experimental intervention study was conducted. We recruited two cohorts of healthy Dutch-speaking women through leaflets and information screens at the prenatal consultation of four hospitals in the province of Limburg. Additional recruitment was carried out through social media during the COVID-19 pandemic.

Recruitment of participants and data collection were carried out in two sequential time periods. The intervention group was recruited between 1 June 2020 and 31 January 2021, allowing for active participation in the intervention. The control group was recruited between 1 February 2021 and 1 September 2021. Participants who expressed interest (*N* = 152) were contacted by e-mail or phone and received complete study information. Women were offered at least a week to consider participation. A reminder was sent after two weeks if no response was received.

### 2.2. Group Assignment

#### 2.2.1. Intervention Group

After obtaining written informed consent, we collected sociodemographic data and assessed eligibility criteria. The women underwent a 15 min telephone interview assessing the Mini International Neuropsychiatric Interview (MINI 5.0) to screen for current psychiatric disorders. Women with indications of major depressive, anxiety or bipolar disorder or expressed suicidal ideations were excluded and advised to contact their general practitioner for referral to specialised mental healthcare. Eligible participants were those who were pregnant, aged ≥ 18 years, not using psychopharmaceuticals and had access to online hardware including a digital device. Women not proficient in Dutch were excluded. Women who met the eligibility criteria were invited to participate in the resilience-enhancing intervention (*N* = 70).

#### 2.2.2. Control Group

Sequential to the intervention group, women were assigned to the control group of the study (*N* = 32). These women did not engage in the resilience-enhancing intervention and received care-as-usual. They completed the same baseline screening and questionnaires as the intervention group participants did at the same moments during pregnancy, postpartum and follow-up (see Table 1). Recruitment and participant flow are illustrated in the flowchart (Figure 1).

#### 2.2.3. Supported Online Resilience-Enhancing Intervention

After inclusion, a baseline measurement (T0) of resilience and resilience attributes was taken (Table 1). All questionnaires were online (LimeSurvey 2.73) and took approximately 20 min to complete. At 28–32 weeks of pregnancy (T1), women were invited to a 1.5 h online group session hosted and moderated by a clinical psychologist. Groups were formed according to gestational age. The content of the group sessions was eclectic, but primarily based on psychoeducation, modes to enhance psychological skills (e.g., coping strategies) and fostering connections among participants. After the first group session, participants were additionally invited to an online platform. Therefore, a closed social learning environment (a Facebook feature) was developed for each group. Distinct from a standard Facebook group, a social learning environment allows the moderator to organize posts, enables participants to indicate completion, and provides insights on unit completion. Through this platform, resilience-enhancing exercises were posted in bi-weekly units. Each unit took approximately two hours (e.g., reading, homework, practicing) to fully complete. Participants had the flexibility to engage in the exercises as little or as much as they wanted each day or week, making it easier to incorporate into their daily schedule. Motivational support quotes were shared weekly on the platform to encourage engagement. The online modules on the platform were permanently available up to one year after childbirth. At 32–36 weeks (T2), an online one-hour group session, co-hosted by a midwife, addressed childbirth-related topics. Topics such as pain management and feeding and caring for the baby after childbirth were explored. In addition, themes such as expectations towards parenthood and the emotional impact on the partner relationship were raised. At three weeks postpartum, participants received a check-in telephone call to ask about their wellbeing and to obtain their consent for the next online questionnaire (T3). The call was made by the psychologist who also coached the group sessions. A final group session was organised at 9–12 weeks postpartum (T4). Afterwards, two follow-up measurements took place at 6 (T5) and 12 months (T6) after childbirth. The control group completed questionnaires at the same fixed timepoints (T0–T6) but did not participate in group sessions, access the Facebook platform or receive a check-in call after childbirth.

### 2.3. Measures

Data were collected via online self-reported questionnaires at baseline (T0), during the intervention (28–32 weeks pregnancy—T1; 32–36 weeks pregnancy—T2; three weeks postpartum—T3), post intervention (9–12 weeks postpartum—T4) and at follow-up, at 6 (T5) and 12 months (T6) after childbirth (Table 1).

#### 2.3.1. Demographic Variables

Women self-reported on age, parity, gestational age, marital status, education level, ethnicity, monthly household income, smoking behaviour and alcohol use at the baseline assessment.

#### 2.3.2. Resilience and Perinatal Resilience Attributes

Resilience was assessed with the validated 25-item Connor–Davidson Resilience Scale (CD-RISC) [25]. The perinatal resilience attributes were assessed using the following:

Social support: The Multidimensional Scale of Perceived Social Support (MSPSS) is a valid 12-item measure to assess perceived social support in pregnant populations [26].

Self-esteem: The Rosenberg Self-Esteem Scale (RSES) is a valid 10-item questionnaire to assess global self-esteem, defined as a person’s overall evaluation of their worthiness as a human being [27].

Self-efficacy: The General Self-Efficacy Scale (GSES) is a valid 10-item questionnaire and used to assess self-efficacy and the sense of personal competence [28].

Sense of mastery: The Five Facet Mindfulness Questionnaire (FFMQ) is a valid 39-item measure to assess mindfulness and sense of mastery in pregnant populations [29].

Given the static nature of personality, this resilience attribute was not included in this study.

#### 2.3.3. Maternal Mental Health

Maternal mental health was measured using the valid 10-item Edinburgh Postpartum Depression Scale (EPDS) [30] to assess symptoms of depression experienced during pregnancy and after childbirth. The State and Trait Anxiety Inventory (STAI) [31] assessed trait and state anxiety in the perinatal period through a valid 40-item questionnaire.

### 2.4. Data-Analysis

To test the potential effectiveness of the supported online resilience-enhancing intervention for pregnant women, participants completed online questionnaires at different timepoints (Table 1). First, descriptive statistics were carried out including distribution examination for all continuous variables. Analyses were conducted to ensure that parametric test assumptions were met. No significant outliers in the variables of interest were found and data distributions were tested with the Shapiro–Wilk test. Independent samples t-tests, Mann–Whitney U-tests and Chi-square tests were used for baseline comparisons between intervention and control group. Correlations were computed using Pearson’s or Spearman’s coefficients depending on data distribution. Mauchly’s Test of Sphericity was used for a repeated-measures analysis of variance; the results indicated that the assumptions of sphericity had not been violated.

A repeated-measures ANOVA examined within-group changes of resilience, resilience attributes and maternal mental health for the intervention and the control group. When the normality assumption was violated, the non-parametric Friedman test was used, followed by post hoc analysis with Wilcoxon signed-rank tests. A repeated-measures ANCOVA was performed to determine whether mean levels of resilience and maternal mental health outcomes differed between the intervention and control group at postintervention (T4) and the 6-month follow-up (T5), controlling for differences in baseline levels. SPSS software (IBM SPSS Statistics for Windows, Version 28.0) was used for analyses, with a *p*-value < 0.05 indicating statistical significance.

## 3. Results

### 3.1. Sample Descriptives

A total of 152 participants were recruited; of these, 102 participants were enrolled in the study. The following reasons were given for non-participation: showing no interest or being too busy (*N* = 46). Four women were excluded based on a positive identification of current mental health problems through the diagnostic interview (MINI 5.0). Figure 1 shows the flowchart of the participants in both intervention (*N* = 70) and control (*N* = 32) group.

There was a significant difference in mean age of women in the intervention and control group (33.7 ± 4.24 years and 29.1 ± 3.49 years, *p* < 0.05). There were no significant differences in terms of marital status, education level, occupation or family income per month between the two groups. There were also no significant differences between the intervention and control group on pregnancy-related characteristics. Table 2 shows general characteristics of the study participants.

Table 3 represents the mean values of dependent variables as well as the standard deviations of the baseline (T0), postintervention (T4) and follow-ups at 6 (T5) and 12 months after childbirth (T6). No statistically significant differences were detected between the two groups at baseline on any variables except for perceived social support (MSPSS) and sense of mastery (FFMQ). The control group (*M* = 79.41, *SD* = 6.0) experienced higher social support (*U* = 768, *p* < 0.05) than the intervention group *M* = 75.60, *SD* = 8.05). Also, the control group had a higher mean sum score (*M* = 132.83, *SD* = 11.05) on the FFMQ (*t*(80.434) = 2.397, *p* = 0.019) than the women in the intervention group (*M* = 125.49, *SD* = 17.07). Internal reliability coefficients were >0.80 for all outcome measures.

### 3.2. Within-Group Changes

#### 3.2.1. Changes in Resilience Scores within the Intervention Group and Control Group

In the intervention group, the mean sum resilience score was relatively stable from baseline (T0) to 12 months postpartum (T6), with a decrease at 6 months postpartum (T5). The within-group changes did not reach statistical significance (*F*(2.34, 77.16) = 1.17, *p* = 0.320) (Figure 2).

In the control group, the mean sum resilience score differed significantly between the four timepoints (*F*(3, 15) = 6.97, *p* = 0.004, *η*^2^ = 0.582) (Figure 2). Post hoc tests using Bonferroni correction revealed that resilience was significantly lower at 12 months postpartum (T6) (*M* = 69.33) than at baseline (T0) (*M* = 75.33), 95% CI 0.235–11.765, *p* = 0.042.

#### 3.2.2. Changes in Resiliency Attributes within the Intervention Group and Control Group

Social support

In the intervention group, a significant difference in perceived social support was found (*χ*^2^(2) = 7.104, *p* = 0.028). Perceived social support was significantly lower at six months follow-up (T5) than at baseline (T0), *Z* = −2.52, *p* = 0.012, *r* = −0.24. There was also a significant difference in perceived social support between postintervention (T4) and follow-up at six months after childbirth (T5), with lower perceived social support at six months after childbirth (T5), *Z* = −2.53, *p* = 0.011, *r* = −0.25. There were no significant differences from baseline (T0) to postintervention (T4) regarding perceived social support (*Z* = −0.73, *p* = 0.463, *r* = −0.07).

There was no statistically significant difference in perceived social support within the control group, (*χ*^2^(2) = 1.289, *p* = 0.547).

Self-esteem

In both the intervention group and the control group, the mean sum scores for self-esteem were relatively stable from baseline (T0) to six months after childbirth (T5). The within-group changes did not reach statistical significance (*F*(2, 78) = 0.505, *p* = 0.605; *F*(2, 42) = 0.810, *p* = 0.452).

Self-efficacy

In both the intervention group and the control group, the mean sum scores for general self-efficacy were relatively stable from baseline (T0) to six months after childbirth (T5). The within-group changes did not reach statistical significance (*F*(2, 90) = 0.515, *p* = 0.599; *F*(2, 46) = 2.924, *p* = 0.064).

Sense of mastery

In the intervention group, the mean sum score for sense of mastery decreased at six months after childbirth (T5) compared to baseline (T0) but did not reach statistical significance (*F*(2, 68) = 0.900, *p* = 0.411).

In the control group, the mean sum score for sense of mastery decreased at postintervention (T4) but increased again at six months after childbirth (T5). The changes, however, did not reach statistical significance (*F*(2, 50) = 2.632, *p* = 0.082).

#### 3.2.3. Changes in Maternal Mental Health within the Intervention and Control Group

Depression

In both the intervention group and the control group, the mean sum scores for depression were relatively stable from second trimester of pregnancy (T1) to six months after childbirth (T5). The within-group changes did not reach statistical significance (*F*(2, 86) = 0.789, *p* = 0.458; *F*(2, 44) = 0.728, *p* = 0.489).

Anxiety

In the intervention group, the mean anxiety score increased from the second trimester of pregnancy (T1) to six months postpartum (T5), but did not reach statistical significance (*t*(44) = −1.769, *p* = 0.084).

In the control group, the mean anxiety score remained stable from the second trimester of pregnancy (T1) to six months postpartum (T5). There was no significant difference over time (*t*(21) = −0.185, *p* = 0.855).

### 3.3. Between-Group Changes: Potential Effect of the Supported Online Resilience-Enhancing Intervention

#### 3.3.1. Does Resilience Increase after Participating in the Intervention?

An independent *t*-test for resilience scores at baseline showed no significant difference between the intervention group and the control group, *t*(74.416) =  1.41, *p* = 0.164. A two-way repeated-measures ANCOVA was conducted to evaluate the difference in resilience scores between the two groups. Age and baseline scores for social support and sense of mastery, were added as covariate variables given the significant difference between the two groups at baseline. There was no significant main effect of time ((*F*(2, 108) = 2.60; *p* = 0.079), nor between the two groups ((*F*(1, 54) = 0.041; *p* = 0.840) (Figure 3).

#### 3.3.2. Do Maternal Mental Health Problems Decrease after Participating in the Intervention?

Depression

An independent *t*-test for depression scores at baseline showed no significant difference between the intervention and control group, *t*(90) =  −0.45, *p* = 0.655. A two-way repeated-measures ANCOVA was conducted to evaluate the differences in depression scores between the two groups. Age and baseline scores for social support and sense of mastery were added as covariate variables given the significant difference between the two groups at the baseline. There was no significant main effect of time ((*F*(1.741, 94.013) = 0.981; *p* = 0.369) nor between the two groups ((*F*(1, 54) = 0.352; *p* = 0.556) (Figure 4).

Anxiety

An independent *t*-test for anxiety scores at baseline showed no significant difference between the intervention and control group, *t*(85) =  −0.951, *p* = 0.344. A one-way repeated-measures ANCOVA was conducted to evaluate the difference in anxiety scores between the two groups. Age and baseline scores for social support and sense of mastery were added as covariate variables given the significant difference between the two groups. There was no significant main effect of time ((*F*(1, 53) = 0.674; *p* = 0.415), nor between the two groups ((*F*(1, 53) = 0.001; *p* = 0.971) (Figure 5).

## 4. Discussion

This paper describes the potential effectiveness of a supported online intervention for pregnant women aimed to enhance resilience and promote maternal mental health. The study compared changes in resilience, resilience attributes and mental health between the intervention and control group. Despite no statistically significant differences between the two groups, interesting within-group trends were observed.

First, resilience remained stable in the intervention group, contrasting with a significant decrease in the control group. All participants in this study faced the COVID-19 pandemic, including the exceptional quarantine measures, social deprivation, fear of infection and concerns around childbirth (e.g., presence of their partner). These additional sources of stress, on top of the challenges associated with future parenthood, could negatively impact the emotional wellbeing of women and put their resilience under pressure [32,33]. In the study of Preis et al. [34], nearly a third of pregnant women experienced elevated levels of stress related to the COVID-19 pandemic [34]. Despite the fact that the results showed no significant increase in resilience, stability in resilience scores within the intervention group may be clinically significant within the COVID-19 context. The meta-analysis of Janitra et al. [35] showed that the prevalence of low resilience in the general population increased from 21% in the period January–March 2020 to 29% in April–June 2020, with a peak of 46% in the period of January–March 2021 [35]. Furthermore, we observe a decline in resilience transitioning from pregnancy to the first year after childbirth, highlighting the impact of childbirth and the challenges associated with this significant life event. Moreover, we see that the 12-week mark after childbirth (T4) represents a crucial point in the first postpartum period. Within the Belgian context, this point coincides with work resumption after maternity leave. This additional stressor can put further pressure on women’s resilience during a period already characterized by many changes and adjustments. Yet, overall resilience levels were relatively high at baseline (intervention group—*M* = 69.09, maximum = 100). Possibly, due to the self-referral recruitment, this intervention attracted the most resilient and motivated women. The use of a healthy, low-risk, sample of pregnant women might provide little room for improvement.

Second, perceived social support was significantly higher in the control group at baseline. This difference could be attributed to recruitment timing, with the intervention group experiencing stricter pandemic restrictions at that time. The recruitment phase for the intervention group coincided with the first and second wave of the COVID-19 pandemic. Quarantine restrictions in Belgium were strong, with a strict lockdown of three months at the start of the pandemic. At the time of recruitment for the control group, social restrictions were less severe and there were more opportunities for support. In addition, it is possible that they who expressed interest in the intervention were looking to strengthen their social support network in times of social restriction, since peer support was one of the main components of the intervention. The decrease in social support between postintervention and follow-up at six months after childbirth in the intervention group might be linked to the ending of the group sessions and reduced activity on the peer support platform. The loss of the feeling ‘we are all in this together’ that prevailed during the group sessions, may have contributed to lower perceived social support [36]. This supports the results of other studies, confirming the strong need to implement social interventions among new parents [23,33,36].

Another important finding of this study is the low attrition rate (19%) compared to other web-based interventions and interventions for treating postpartum depression in primary care [37,38]. At postintervention, 81% participants were still actively involved in the intervention. A possible explanation may be that the usual perinatal care services were limited due to the COVID-19 measures. Additionally, the online format and the human support approach may contribute to the high adherence rate. Pregnant women find online interventions acceptable and appealing [12,39,40,41]. Health-related apps or online sources with information related to physical health in pregnancy, foetal development and practical aspects of the transition to parenthood are widely used and frequently accessed by pregnant women [42]. However, studies on online interventions focusing on the psychological and social aspects of (future) parenthood are limited [43,44]. Participants in this study reported that the online format facilitated participation, encouraged them to share experience and fostered a sense of openness due to anonymity. The developed intervention incorporated a large human support component, designed to stimulate peer support and foster interaction between participants and researchers (psychologist and midwife). Supported web-based interventions focusing on perinatal mental health may be promising approaches in the prevention of MMHPs [20,45].

Additionally, the intervention was specifically designed for expectant mothers incorporating multiple components: resilience-enhancing exercises, online group sessions and a peer support platform. These components were selected based on the needs of mothers whose resilience was under pressure during pregnancy and the first year after childbirth [46]. Ayers et al. [47] stated that intervention research needs to move away from a ‘one-size fits all’ approach [47]. Therefore, the developed intervention in this study combines a range of different strategies to offer a personalised approach tailored to the needs of the participants. This may also be an explaining factor for the high adherence rate within this study. Giving the preventive approach, the intervention was designed to promote mental health rather than to reduce existing symptoms. This might explain the nonsignificant findings on maternal mental health outcomes.

### 4.1. Strengths and Limitations

A first strength of this study is its innovative focus on resilience attributes using a longitudinal design and directly assessing resilience through the CD-RISC. In contrast to prior research, this study broadens the perspective on perinatal mental health by not only investigating negative outcomes (e.g., depression) but also examining positive outcomes such as resilience and resilience attributes. A second strength is the intervention’s thorough development process [21], which is based on the perinatal resilience model [18] and informed by the BCW framework [19], preceded this pilot study. Third, the accessibility, online delivery method, and easily applicable nature of the intervention, either as a whole or based on the individual components, increase the potential for widespread implementation.

However, certain limitations need to be considered. First, recruitment relied mainly on individual’s motivation and most participants entered the study through self-referral. This may have led to a relatively homogeneous sample of Caucasian, well-educated women with widespread access to technology which limits the generalizability of the findings [48]. Second, the lack of randomization induced the risk of bias due to the unequal distribution of confounders between the groups. Additionally, there is a potential for bias when considering the impact of the pandemic on the intervention and control groups, given their distinct circumstances during this period. The COVID-19 restrictions changed during the data collection period, varying from strict lockdowns to more lenient rules. This variability holds the potential to influence the experiences of participants during and after pregnancy and childbirth, particularly considering the sequential recruitment of the intervention and control groups. Consequently, it may have implications for outcome measures (e.g., perceived social support). Furthermore, participation intensity (e.g. uptake of exercises) was not registered and thus not controlled for in the analysis. Another limitation is that we exclusively relied on self-reported measures. It is acknowledged that there was a risk of participant fatigue with the number of questionnaires. However, it was anticipated that the online survey would take no longer than 15–20 min in total to complete. Unfortunately, a technical malfunction led to incomplete control group follow-up data at 12 months after childbirth (T6). At last, given the exploratory nature of this pilot study testing a novel intervention, a power analysis was not conducted. Sample size determination was driven by pragmatic considerations, potentially resulting in a sample size that might be insufficient to detect a clinically significant difference.

### 4.2. Recommendations for Future Research

Future research needs to evaluate whether supported web-based interventions are acceptable and effective for pregnant women from other ethnicities and other education groups, who may be less likely to self-refer and interact differently with online services and interventions. Another party that is currently missing in perinatal mental health research is the partner. Further research on the needs and experiences of fathers, co-mothers or other parenting dyads would be of value to include. Within future studies, a randomized controlled trial is preferable in further investigating the effects of resilience and resilience attributes on the mental health of (expectant) parents. Also, adding biological measures (e.g., heart rate variability) may be interesting to include for validation measures. An adaptation of the current intervention in which the framework of Acceptance and Commitment Therapy (ACT) is included may be interesting. This approach is increasingly popular for the prevention and treatment of perinatal mental health problems and comprises resilience factors such as psychological flexibility and mindfulness.

## 5. Conclusions

This paper contributes to the literature on maternal mental health by highlighting the potential value of a supported online intervention comprising different deployable components: online group sessions, resilience-enhancing exercises and an online peer support platform. The intervention was designed to stimulate human support through online interaction and peer support. The human-supported approach combined with the online format is intended to address potential shortcomings and enhance the overall effectiveness of online prenatal interventions. Moreover, the study highlights the importance of support, not only during pregnancy but also in the first year after childbirth. A larger-scaled, randomized–controlled trial to test the effectiveness in everyday practice is recommended.

## Figures and Tables

**Figure 1 ijerph-21-00209-f001:**
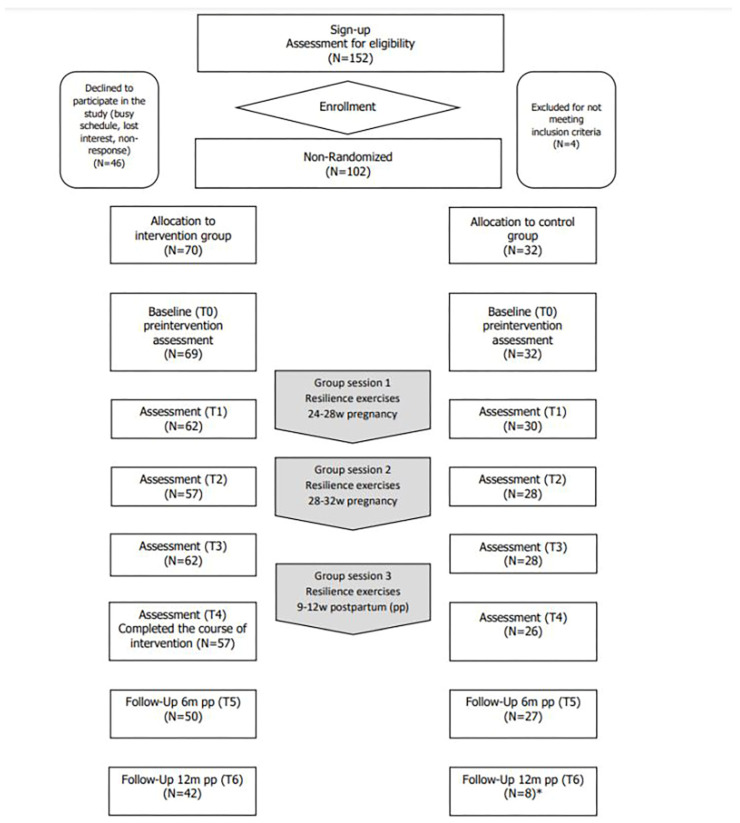
Flowchart of enrolment, interventions and assessments. Note: * due to technical problems with the survey program, only 8 questionnaires were registered for the second follow-up measurement (12 months postpartum) in the control group.

**Figure 2 ijerph-21-00209-f002:**
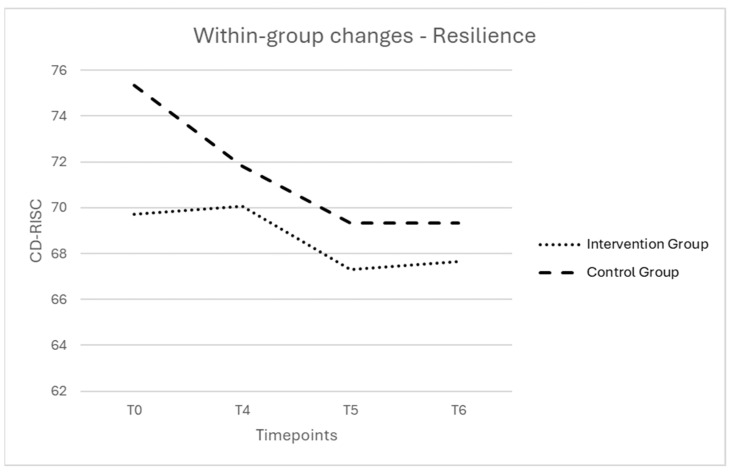
Evolution during the study period (T0–T6) of the estimated marginal means of resilience is presented for the intervention group and the control group. Note. T0 = baseline; T4 = postintervention; T5 = follow-up 6 months after childbirth; T6 = follow-up 12 months after childbirth; CD-RISC = Connor–Davidson Resilience Scale (range 0–100).

**Figure 3 ijerph-21-00209-f003:**
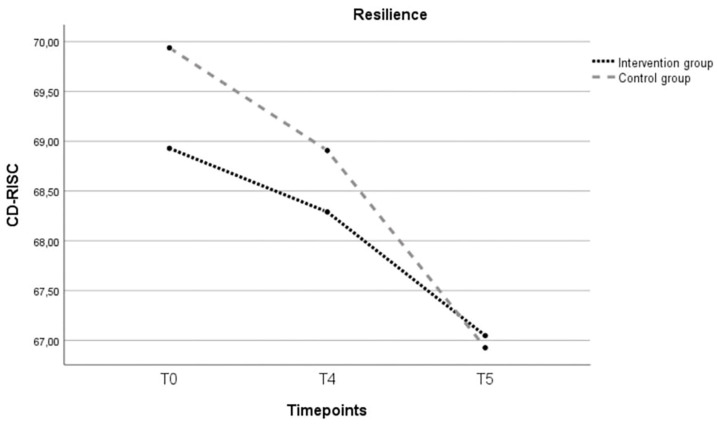
Evolution over three timepoints (T0, T4, T5) of the estimated marginal mean sum resilience score for the intervention and control group. Note: T0 = baseline; T4 = postintervention; T5 = follow-up 6 months after childbirth; CD-RISC = Connor–Davidson Resilience Scale (range 0–100); covariates appearing in the model are age = 31.9492; baseline score sense of mastery = 127.729; baseline score perceived social support = 76.339.

**Figure 4 ijerph-21-00209-f004:**
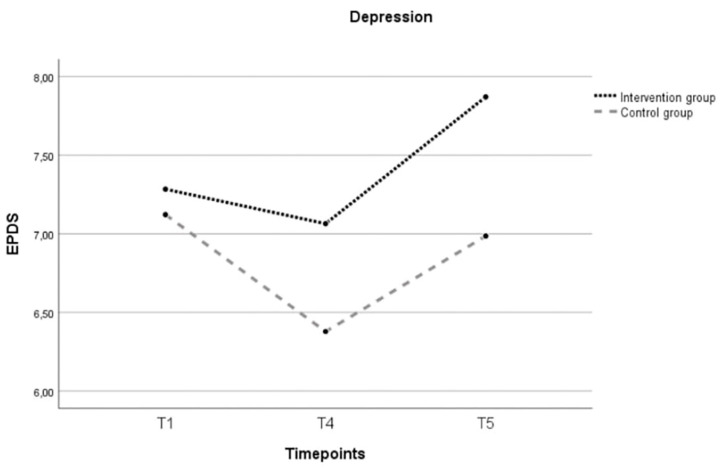
Evolution over three timepoints (T0, T4, T5) of the estimated marginal mean sum depression score for the intervention and control group. Note: T1 = 28–32 weeks of pregnancy; T4 = postintervention; T5 = follow-up 6 months after childbirth; EPDS = Edinburgh Postnatal Depression Scale (range 0–30); covariates appearing in the model are age = 31.9492; baseline score sense of mastery = 127.729; baseline score perceived social support = 76.339.

**Figure 5 ijerph-21-00209-f005:**
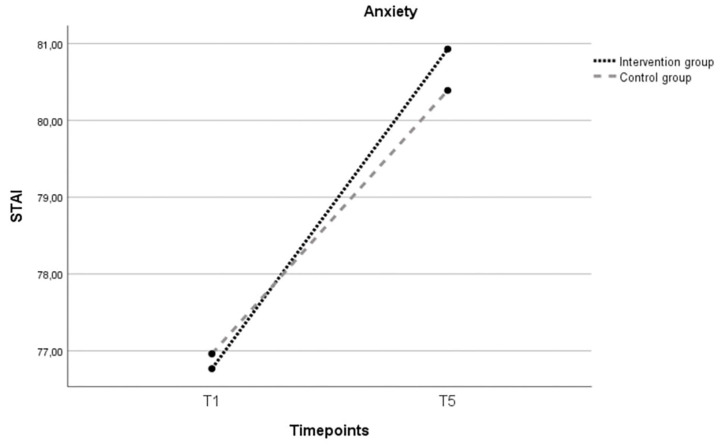
Evolution over three timepoints (T0, T4, T5) of the estimated marginal mean sum anxiety score for the intervention and control group. Note: T1 = 28–32 weeks of pregnancy; T5 = follow-up 6 months after childbirth; STAI = State Trait Anxiety Inventory (range 20–80); covariates appearing in the model are age = 31.9492; baseline score sense of mastery = 127.729; baseline score perceived social support = 76.339.

**Table 1 ijerph-21-00209-t001:** Timings of assessments.

Scales	T0 (Baseline)	T1 (28–32 w)	T2 (32–36 w)	T3 (3 w pp)	T4 (12 w pp)	T5 (6 m pp)	T6 (12 m pp)
CD-RISC	X		X	X	X	X	X
MSPSS	X				X	X	
RSES	X				X	X	
GSES	X				X	X	
FFMQ	X				X	X	
EPDS		X			X	X	
STAI		X				X	

Note. CD-RISC = Connor–Davidson Resilience Scale (25 items); MSPSS = Multidimensional Scale of Perceived Social Support (12 items); RSES = Rosenberg Self-Esteem Scale (10 items); GSES = General Self-Efficacy Scale (10 items); FFMQ = Five-Facet Mindfulness Questionnaire (39 items); EPDS = Edinburgh Postpartum Depression Scale (10 items); STAI = State and Trait Anxiety Questionnaire (40 items); T = timepoint; w = weeks; pp = postpartum.

**Table 2 ijerph-21-00209-t002:** Description of sociodemographic and pregnancy-related characteristics assessed at T0.

	Intervention Group (*N* = 70)	Control Group (*N* = 32)	Group Differences (*p*-Value)
*Sociodemographic Characteristics*			
**Age in years Mean (SD) ***	33.7 (4.24)	29.1 (3.49)	0.00
**Marital status (%)**			
Married or legally cohabiting	97.1	93.8	
Single	2.9	6.3	
**Education level (%)**			0.683
Secondary education	8.6	6.3	
Bachelor’s degree	44.3	37.5	
Master’s degree	47.1	56.3	
**Occupation (%)**			0.202
Employee	50.0	68.8	
Civil servant	24.3	15.6	
Self-employment	10.0	6.3	
Pregnancy leave	5.7	9.4	
**Family income/month (%)**			0.853
EUR 1500–2000	1.4	3.1	
EUR 2000–3000	5.7	12.5	
EUR 3000–4000	44.3	40.6	
EUR 4000–5000	37.1	31.3	
>5000 EUR	8.6	9.4	
Unknown	2.8	3.1	
**Nationality**			1.000
Belgian	98.6	96.9	
Other Western country	1.4	3.1	
*Pregnancy-related characteristics*			
**Parity (%)**			0.813
Primiparous	77.1	75.0	
Multiparous	22.9	25.0	
**Method of conception**			0.569
Spontaneous	87.1	90.6	
Hormone treatment	5.7	6.3	
IVF/ICSI	7.1	3.1	
**History of miscarriage %**			1.000
No	88.6	90.6	
Yes	11.4	9.4	
**Alcohol use** during pregnancy	0.0	0.0	
**Smoking** during pregnancy	0.0	0.0	

Note. T0 = baseline. * Age differed significantly between the intervention and control group (*p* < 0.05).

**Table 3 ijerph-21-00209-t003:** Overview of mean values and standard deviations of dependent variables at T0, T2, T3, T4, T5, and T6.

	Intervention Group											
Construct	Baseline (T0)		32–36 w (T2)		3 w pp (T3)		Post Intervention (T4)		Follow-up 6 m pp (T5)		Follow-up 12 m pp (T6)	
	*M*	*SD*	*M*	*SD*	*M*	*SD*	*M*	*SD*	*M*	*SD*	*M*	*SD*
Resilience	69.09	12.36	68.61	11.47	69.70	11.09	69.02	10.93	67.02	13.32	68.09	13.69
	*N* = 65		*N* = 57		*N* = 62		*N* = 57		*N* = 50		*N* = 42	
Social support	75.60	8.05					75.72	8.57	73.64	9.96		
	*N* = 69						*N* = 57		*N* = 53			
Self-esteem	20.80	4.32					21.18	4.61	20.55	4.89		
	*N* = 57						*N* = 54		*N* = 53			
General self-efficacy	30.52	4.38					30.48	3.94	30.11	4.87		
	*N* = 67						*N* = 57		*N* = 53			
Sense of mastery	125.49	17.07					123.59	16.86	124.56	18.50		
	*N* = 55						*N* = 49		*N* = 53			
	T1						T4		T5		T6	
Depression	7.26	4.18					7.66	4.87	8.43	5.31		
	*N* = 62						*N* = 56		*N* = *53*			
Anxiety	76.94	21.13							82.53	22.21		
	*N* = 61								*N* = 49			
Stait anxiety	38.44	12.34							41.09	12.61		
	*N* = 62								*N* = 52			
Trait anxiety	38.73	9.53							40.64	10.45		
	*N* = 61								*N* = 51			
	**Control Group**											
**Construct**	**Baseline** **(T0)**		**32–36 w** **(T2)**		**3 w pp (T3)**		**Post Intervention (T4)**		**Follow-up 6 m pp** **(T5)**		**Follow-up 12 m pp** **(T6)**	
	*M*	*SD*	*M*	*SD*	*M*	*SD*	*M*	*SD*	*M*	*SD*	*M*	*SD*
Resilience	72.17	8.31	71.53	10.27	69.71	10.75	71.42	11.04	69.85	10.58	69.37	5.29
	*N* = 28		*N* = 28		*N* = 28		*N* = 26		*N* = 27		*N* = 8 *	
Social support	79.41	6.00					78.40	5.98	79.18	5.60		
	*N* = 31						*N* = 57		*N* = 25			
Self-esteem	22.06	4.97					22.46	4.38	22.18	4.91		
	*N* = 30						*N* = 54		*N* = 24			
General self-efficacy	31.90	3.75					31.04	3.40	31.50	4.13		
	*N* = 31						*N* = 57		*N* = 26			
Sense of mastery	132.83	11.06					129.51	11.64	130.92	11.82		
	*N* = 30						*N* = 49		*N* = 27			
	T1						T4		T4		T5	
Depression	6.83	4.43					5.77	4.10	6.27	3.69		
	*N* = 30						*N* = 56		*N* = 26			
Anxiety	72.50	16.79							74.85	16.09		
	*N* = 26											
Stait anxiety	35.77	9.12							38.38	9.45		
	*N* = 27								*-*		*N* = 26	
Trait anxiety	36.50	8.23							36.46	8.04		
	*N* = 28								*N* = 26			

Note. M = mean; SD = standard deviation; T = timepoint; w = weeks; m = months; pp = postpartum. * due to technical problems with the survey program, only 8 questionnaires were registered for the second follow-up measurement (12 months postpartum) in the control group.

## Data Availability

The anonymized dataset and syntaxes can be obtained for research interests by reasonable request from the corresponding author.

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
