# Peer review of "A Supported Online Resilience-Enhancing Intervention for Pregnant Women: A Non-Randomized Pilot Study"

_ijerph, 2024, doi:10.3390/ijerph21020209_

Round 1
Reviewer 1 Report
Comments and Suggestions for Authors
Authors developed a supported online resilience-enhancing intervention for pregnant women. They reported that though resilience score of both intervention and control decreased after childbirth, the intervention group demonstrated stable resilience.
It is very important to take care of mental health of pregnant women. Since women with a baby have difficulty in participating meetings in-person, online program is preferred.
I have a minor concerns about the manuscript.
1. From Table 1, they assessed CD-RISC at T0, T2, T3, T4, T5, and T6. However, In Table 3 and Figure 2, they did not report the score of T2 and T3. T2 is just before childbirth and T3 is 3 weeks after childbirth and I feel that they have so much difficulties especially at T3. Please show the scores at T2 and T3.
2. Authors showed timecourse of CD-RISC separately in Figure 2. It might be better to put the graphs in the same graph like Figure 3. At the same time, I see that the scores are differetn between Figure 2 and Figure 3; Though the CD-RISC of control at T0 is around 75 in Figure 2(a), it is around 70 in Figure 3. The scores at Table 3 are different from both graphs. Please check if there are any mistakes in the table or figures.
3. From Table 2, I see that around 23-25% of participants are multiparous. Is there any difference between primiparous and multiparous? I understand that sub-analysis might be difficult because of small sample size, I would like to know if they see any difference between them.
4. I am a little bit surprised to see that resilience in the control group did not fully recover at T6. Are there any reason why their resilience did not recover beside COVID-19?
Author Response
The research team would like to thank the reviewers for their feedback on this manuscript. Due to the adjustments, we hope we have been able to optimise the manuscript.

Reviewer 2 Report
Comments and Suggestions for Authors
Congratulations on your hard work and the sound and appealing presentation of your results!
I would like to challenge you to be more 'self'-critical in your discussion of the need for such interventions, the way in which this pilot study has implemented it, and its outcomes
As you mention, these kind of approaches are increasingly popular, but I am not convinced by the evidence you portray, that the science supports this popularity
Author Response

(The authors gave the same response as above.)

Reviewer 3 Report
Comments and Suggestions for Authors
It is important to detail that the lack of randomization could induced the risk of bias not only due to the unequal distribution of confounders between the groups (as they detail in the Strengths and Limitations apart but also due to the different timing of follow - up of each group
Thus, we could be finding important biases derived from the moment of the pandemic in which the intervention group was with respect to the moment furthest from the pandemic in which the study was carried out in the control group.
Author Response

(The authors gave the same response as above.)

Reviewer 4 Report
Comments and Suggestions for Authors
This is an Internet intervention study focused on enhancing resilience and promoting maternal mental health of pregnant women.
1. Check https://www.mdpi.com/journal/ijerph/instructions - "The abstract should be a total of about 200 words maximum. The abstract should be a single paragraph and should follow the style of structured abstracts, but without headings".
2. What do the authors mean by "The intervention is relatively ''brief''"?
3. Improve the resolution/quality of all figures in the manuscript. They are hard to read.
4. References for questionnaires and scales should be provided (sections 3.2 and 3.3).
5. Section 3. Measures should be inside section 2. Materials and Methods.
6. [Main concern]: The authors developed an intervention: "an online platform which was developed as a closed social learning environment (Facebook)". However, very little information is provided by the authors about this platform. Description and images/screenshots could be provided in the manuscript or as supplementary material.
Author Response

(The authors gave the same response as above.)
